# Toxicity and Synergistic Effect of *Elsholtzia ciliata* Essential Oil and Its Main Components against the Adult and Larval Stages of *Tribolium castaneum*

**DOI:** 10.3390/foods9030345

**Published:** 2020-03-16

**Authors:** Jun-Yu Liang, Jie Xu, Ying-Ying Yang, Ya-Zhou Shao, Feng Zhou, Jun-Long Wang

**Affiliations:** School of Life Science, Northwest Normal University, Lanzhou 730070, Gansu, China; XJ13893279431@163.com (J.X.); y17339832389@163.com (Y.-Y.Y.); 18893811951@163.com (Y.-Z.S.); fengzhou@nwnu.edu.cn (F.Z.); wangjunlong@nwnu.edu.cn (J.-L.W.)

**Keywords:** *Elsholtzia ciliata*, *Tribolium castaneum*, essential oil, carvone, limonene, insecticidal activity, synergistic effect

## Abstract

Investigations have indicated that storage pests pose a great threat to global food security by damaging food crops and other food products derived from plants. Essential oils are proven to have significant effects on a large number of stored grain insects. This study evaluated the contact toxicity and fumigant activity of the essential oil extract from the aerial parts of *Elsholtzia ciliata* and its two major biochemical components against adults and larvae of the food storage pest beetle *Tribolium castaneum*. Gas chromatography–mass spectrometry analysis revealed 16 different components derived from the essential oil of *E. ciliata*, which included carvone (31.63%), limonene (22.05%), and *α*-caryophyllene (15.47%). Contact toxicity assay showed that the essential oil extract exhibited a microgram-level of killing activity against *T. castaneum* adults (lethal dose 50 (LD_50_) = 7.79 μg/adult) and larvae (LD_50_ = 24.87 μg/larva). Fumigant toxicity assay showed LD_50_ of 11.61 mg/L air for adults and 8.73 mg/L air for larvae. Carvone and limonene also exhibited various levels of bioactivity. A binary mixture (2:6) of carvone and limonene displayed obvious contact toxicity against *T. castaneum* adults (LD_50_ = 10.84 μg/adult) and larvae (LD_50_ = 30.62 μg/larva). Furthermore, carvone and limonene exhibited synergistic fumigant activity against *T. castaneum* larvae at a 1:7 ratio. Altogether, our results suggest that *E. ciliata* essential oil and its two monomers have a potential application value to eliminate *T. castaneum*.

## 1. Introduction

Food security has always been a staple of discussion. Investigations have indicated that insects pose a great threat to global food security by damaging food crops and other food products derived from plants [1]. However, several pests show resistance, and the utilization of existing insecticides has more or less some side effects. For example, many of them can be lethal to nontarget organisms, and the residues of insecticides in crops also have negative impacts on human beings and the environment [2,3]. *Tribolium castaneum* is a species of beetle that is considered as a worldwide pest affecting mainly stored food products, such as grains, flour, and cereals, among others. These are dominant populations of insects found in stored traditional Chinese medicines [4]. *T. castaneum* can damage a great range of food and processed products, leading to agglomeration, discoloration, and spoilage, which result in serious economic losses [5]. The principal method to control these insects is the use of synthetic insecticides or fumigants. However, these methods may cause health hazards to warm-blooded animals, lead to environmental pollution, and potentially bring about insecticide-resistant insects, resulting in pest resurgence [6]. When dealing with food storage and preserving cultural relics and archives, it is essential to not only protect these materials from pests but to also reduce the extent of pesticide residues and avoid pollution. Therefore, an increasing number of researchers are searching and investigating different active natural products as botanical insecticides [7,8].

The essential oils extracted from various plants exhibit unique botanical and medicinal uses that, upon proper application, may not cause detrimental effects in humans and animal health as well as the environment. Essential oils are proven to have significant effects against a large number of stored grain insects, acting through ingestion [9] and contact toxicity [10,11]. The modes of action of plant essential oils on pests may include contact toxicity, fumigant, antifeedant, repellent, and growth-inhibiting activities [12,13]. Essential oils and their constituents from many plants have previously been confirmed to contain insecticidal or repellent activity, which inhibit the growth of insects that damage stored products [14,15,16]. Plant essential oils are often complex mixtures of terpenoids, and their bioactivity is likely to frequently be a result of synergy among constituents [17]. In addition, essential oils and their mono- and sesquiterpenoid constituents are fast-acting neurotoxins in insects, possibly interacting with multiple types of receptors [18]. Research has shown that, for rosemary (*Rosemarinus officinalis*) and lemongrass (*Cymbopogon citratus*) oils, synergy among major constituents results from increased penetration of toxicants through the insect’s integument rather than through inhibition of detoxicative enzymes [19,20]. Moreover, these essential oils are volatile, and the products are also not risky for other organisms [21].

*Elsholtzia ciliata* (Thunb.) Hyland is a widely spread plant in China and is part of the herbal medicine collection with distinct special aroma [22,23,24,25]. The essential oils of *Elsholtzia* have certain poisonous activity on a variety of storage pests [26]. The *E. ciliata* essential oil was found to possess fumigant toxicity and contact toxicity against *Liposcelis bostrychophila*, with a lethal dose 50 (LC_50_) value of 475.2 μg/L and 145.5 μg/cm^2^, respectively [27]. The ether extract of *Elsholtzia stauntonii* had a strong fumigation effect on adult *Sitophilus zeamais* and *T. castaneum*. After four days of treatment, the adult mortality of *S. zeamais* reached over 95%, while it reached 100% for *T. castaneum* [28]. However, a literature survey showed no reports on insecticidal activity of the essential oil from the aerial parts of *E. ciliata* against *T. castaneum*. The present study was therefore undertaken to investigate the chemical components and insecticidal activities of the essential oil, including its active biochemical constituents against the food storage pest *T. castaneum*.

Carvone is a component of caraway (*Carum carvi Linnaeus*), dill *(Anethum graveolens Linnaeus*), and spearmint (*Mentha spicata Linnaeus*) seeds [29]. It is widely used in pesticides, food flavoring, feed flavoring, feed additive, personal care products, and veterinary medicine [30]. Limonene is listed in the Code of Federal Regulations as a generally recognized as safe (GRAS) substance for flavoring agents. It is commonly used in food items, such as fruit juices, soft drinks, baked goods, ice cream, and pudding [31], and it can be directly used in perfumes. It is also used in many flavor formulas with safety amount up to 30%, and the International Fragrance Association (IFRA) has no restrictions on it [32], although the potential occurrence of skin irritation necessitates regulation of this chemical as an ingredient in cosmetics. In conclusion, the use of limonene in cosmetics is safe under the current regulatory guidelines for cosmetics [33,34].

A literature survey showed some reports on insecticidal activity of carvone and limonene against insects. For instance, Fang et al. [35] stated that carvone and limonene had contact toxicity against *Sitophilus zeamais* with LD_50_ values of 2.79 μg/adult and 29.86 μg/adult, respectively. Carvone and limonene also possessed strong fumigant toxicity against *S. zeamais* (LC_50_ = 2.76 and 48.18 mg/L). Yang [36] found that, after 24 h exposure time, the mortalities of insects in carvone with three fumigant concentrations reached 100%. In addition, the limonene showed contact toxicity against *T. castaneum* adults with a LD_50_ value of 14.97 μg/adult [37].

## 2. Materials and Methods

### 2.1. Plant Materials and Extraction of Essential Oil

*E. ciliata* was gathered in Longxi County (35°1′ N latitude, 104°27′ E longitude, altitude 1880 m) in the Gansu province of China. To obtain the crude essential oil, the minced sample was connected to the distillation unit and condenser and maintained for 6 h. Anhydrous Na_2_SO_4_ was added to the crude essential oil to remove all water residue. The volume of the pure essential oil was recorded and the yield was calculated. The prepared essential oil was stored in the refrigerator at 4 °C until use.

### 2.2. Test Insects

*T. castaneum* adults were inoculated into a mixture of whole wheat flour and yeast flour at a mass ratio of 10:1 and cultured in a constant temperature incubator at 30 ± 1 °C with 75% ± 5% relative humidity for 24 h dark treatment. All adult beetles used in the experiment were considered as adult stage after an eclosion time of 1–2 weeks. On the other hand, the test larvae [38] were six instar larvae with an approximate length of 5–6 mm.

### 2.3. Gas Chromatography-Mass Spectrometry (GC-MS) Analysis

The GC-MS analysis was run on an Agilent 6890 N gas chromatograph connected to an Agilent 5973 N mass selective detector. They were equipped with a gas chromatography-flame ionization detector (GC-FID) and a HP-5MS (30 cm × 0.25 mm × 0.25 µm) capillary column. The essential oil sample was diluted in *n*-hexane to obtain a 1% solution. The injector temperature was maintained at 250 °C with the volume injected being 1 µL. The flow rate of carrier gas (helium) was 1.0 mL/min, with the mass spectra scanned from 50 to 550 *m*/*z*. 

The retention indices (RI) were determined from gas chromatograms using a series of *n*-alkanes (C_5_-C_36_) under the same operating conditions. Based on RI, the chemical constituents were identified by comparing them with *n*-alkanes as a reference. The components of the essential oil were identified by matching their mass spectra with various computer libraries (Wiley 275 libraries, NIST 05, and RI from other literature) [39]. 

### 2.4. Contact Toxicity

The contact toxicity activities of *E. ciliata* essential oil and its main components were determined by the dot contact method [40]. The essential oil was diluted to five different concentration gradients (5%, 3.3%, 2.2%, 1.48%, 0.98%) with *n*-hexane. A 0.5 μL diluted solution was dropped on the torso of *T. castaneum* after being palsied by the freezing method. Then, the test insects were transferred to a glass bottle with a volume of 25 mL. *n*-Hexane and pyrethrin were used as negative and positive controls, respectively. Each concentration was repeated 5 times, and 10 test insects were used for each assay. After 24 h, the number of dead insects was recorded, and the mortality and corrected mortality were calculated. Insects that did not respond to a brush were considered dead. A similar experimental method was undertaken in testing the larval stage.

### 2.5. Fumigant Toxicity

Fumigant activities of *E. ciliata* essential oil and its main components against adults and larvae of *T. castaneum* were evaluated based on the method described by Wu et al. [41]. The essential oil was diluted with *n*-hexane to obtain five concentration gradients (10%, 6.6%, 4.4%, 2.9%, 1.77%). Diluted liquids of 10 μL were injected on the filter paper (2.0 cm^2^) and placed on the inside of the bottle cap. The bottle cap was quickly screwed up and wrapped by the sealing film to form a closed space after 20 s. *n*-Hexane was used as a negative control, whereas methyl bromide and phoxim were used as positive controls for adults and larvae, respectively. Each concentration was repeated 5 times and tested in 10 test insects in each assay. After 24 h, the death of the test insects was observed and recorded, and the mortality and corrected mortality were calculated. The same experimental method was used to test the larval stage.

### 2.6. Two Main Components Compounding

We used the ten-point theory [42] that assumes that the half-lethal concentrations of A and B are determined by the virulence of a and b. Hence, the A + B mixture was evaluated by the co-toxic factor method. A total of 7 ratios were selected according to the corresponding concentration gradient order of 1:7, 2:6, 3:5, 4 4, 5:3, 6:2, and 7:1. The contact toxicity and fumigant toxicity methods were performed as described previously (Materials and Methods Section 2.4 and Section 2.5). Three repetitions were done for each treatment, and a blank control was set.

### 2.7. Data Analysis

The LC_50_ (mg/L air) and the LD_50_ (μg/adult or larva) of the lethal activity were analyzed and calculated using SPSS 22.0 statistical software, and the corrected mortality was calculated by Abbott’s formula. The determination of the synergistic effect was performed with combined toxicity evaluation using Sun Yunpei’s co-toxicity method CTC (Co-toxicity index) [43]. The criteria were as follows: 80 ≤ CTC ≤ 120 indicated an additive effect, CTC > 120 indicated a synergistic effect, and CTC < 80 indicated an antagonistic effect. The calculations were as follows:
①Co-toxicity index (CTC) = ATI/TTI × 100%②Mixed virulence index (ATI) = standard drug LD_50_/mixture (A+B) LD_50_ × 100%③Theoretical virulence index of (A+B) (TTI) = Va × Ma + Vb × MbV_a_ = Virulence index of agent A, M_a_ = the mass fraction of agent A in the mixtureV_b_ = Virulence index of agent B, M_b_ = the mass fraction of agent B in the mixture④Single dose virulence index (TI) = standard drug LD_50_/LD_50_ for the test agent × 100%

### 2.8. Chemicals

Pyrethrins were purchased from Dr. Ehrenstorfer GmbH, Augsburg, Germany with a concentration of 27%. Phoxim were purchased from Dr. Ehrenstorfer GmbH, Augsburg, Germany with a purity of 98.0%; Carvone was purchased from Tishila (Shanghai) Chemical Industry Development Co., Ltd., China, with a purity of 99.0%. Limonene was purchased from Shanghai Aladdin Biochemical Technology Co., Ltd., China, with a purity of 95.0%.

## 3. Results

### 3.1. Chemical Compounds of E. ciliata Essential Oil

The essential oil extracted from the leaves of *E. ciliata* had a yield of 0.36% (*V*/m). The chemical compounds and relative contents of *E. ciliata* essential oil are shown in Table 1. In this study, we identified 16 compounds in *E. ciliata* essential oil, the main compounds were monoterpenoids and sesquiterpenes, with monoterpenoids accounted for 76.97%, sesquiterpenes accounted for 20.61%, and carvone was the highest monoterpenoid among all, while *α*-caryophyllene had the highest content of sesquiterpenes. What is more, we observed four major components of *E. ciliata* essential oil, namely, carvone (31.6%), limonene (22.05%), *α*-caryophyllene (15.47%), and dehydroelsholtzia ketone (14.86%). These components are distinct from previous works. For example, *E. ciliata* essential oil derived from Mao’er Mountain of northeastern China mainly constituted dehydroelsholtzia ketone (68.35%) and elsholtzia ketone (25.19%) [44]. More than 30 components were separated from the essential oil of *E. ciliata* in Changbai Mountains in northeastern China, and the main components were *β*-dehydrogeranione (51.77%) and elsholtzia ketone (33.33%) [45]. In addition, the elsholtzia ketone concentration in the essential oil from both Changbai Mountains and Mao’er Mountain in the Liu’s research was higher than that in this experiment. The dehydroelsholtzia ketone in the essential oil from Mao’er Mountain (68.35%) in Liu’s research was double that of this experiment. All the *E. ciliata* in the abovementioned works were gathered from Northeast China, while the *E. ciliata* studied in this paper was from Northwest China. The large climate difference between the two areas may be one of the reasons for the differences in essential oil composition. Moreover, the difference in harvesting time and growth years may also cause differences in essential oil components. 

### 3.2. Contact Activity

Table 2 shows the results of contact activities of *E. ciliata* essential oil and the two main components (carvone and limonene) against *T. castaneum* adults and larvae. The essential oil of *E. ciliata* showed obvious contact toxicity against *T. castaneum* adult and larval stages with LD_50_ of 7.79 μg/adult and 24.87 μg/larva, respectively. Among the two main components, carvone had stronger contact activity against adults (LD_50_ = 5.08 μg/adult), which was 7.59-fold higher than the effect of limonene (LD_50_ = 38.57 μg/adult). This result implies that carvone might have been a key component of *E. ciliata* essential oil involved in contact toxicity against *T. castaneum*. Although the contact activities of essential oil and carvone against *T. castaneum* adults was weaker than that of the positive control pyrethrin (LD_50_ = 0.09 μg/adult), the *E. ciliata* essential oil showed stronger contact effect than previously reported plants. For example, Wu et al. [41] found that the LD_50_ of *Platycladus orientalis* essential oil against *T. castaneum* was 48.59 μg/adult. The essential oils of *Murraya exotica* aerial parts showed contact toxicity against *T. castaneum* adults with LD_50_ values of 20.94 μg/adult [46]. Therefore, *E. ciliata* essential oil and its two main components (carvone and limonene) have strong contact toxicity against *T. castaneum*.

### 3.3. Fumigation Activity

Fumigation activity of *E. ciliata* essential oil and its two components are shown in Table 3. Both *E. ciliata* essential oil and the two major components had obvious fumigant toxicity against *T. castaneum* adults and larvae, although *E. ciliata* essential oil had a stronger fumigating effect on *T. castaneum* larvae (LC_50_ = 8.73 mg/L air). The fumigant toxicity of carvone against adults (LC_50_ = 4.34 mg/L air) was significantly higher than that against larvae (LC_50_ = 28.71 mg/L air). Limonene also had obvious fumigation activity against adults, with a LC_50_ of 5.52 mg/L air. The fumigation effect of carvone and limonene was 2.68 and 2.1 times greater, respectively, than the effect of the essential oil against adults. When the two components were applied together, the fumigation activity increased significantly. A previous study has also reported that carvone and limonene have strong fumigation activity against *T. castaneum* [36]. Therefore, it can be inferred that carvone and limonene are two of the active ingredients containing fumigant toxicity against *T. castaneum*.

For the fumigation effect against larvae, *E. ciliata* essential oil had the best fumigation activity, which was 3.29 times higher than the effect of carvone and 2.36 times higher than that of limonene. The fumigation activity of *E. ciliata* essential oil and the two components appeared weak. The fumigation activity of essential oil was 6-fold weaker than the positive control, and the fumigation activities of carvone and limonene against *T. castaneum* adults was weaker than methyl bromide. However, compared with the fumigation effect of other essential oils, *E. ciliata* essential oil and the two monomers had relatively stronger activity. For instance, Han et al. [47] reported eugenol had contact toxicity against *T. castaneum* larvae and adults with LC_50_ values of 219.00 μL/mL and 363.08 μL/mL, respectively. In addition, Lv et al. [48] used Soxhlet extraction and ether as a solvent to extract essential oils from garlic, chili powder, citrus peel, and toon bark, which showed fumigation activity against *T. castaneum* larvae but not against adults. Given the characteristic of *E. ciliata* essential oil, it is most likely to develop a fumigant insecticide effect against the larvae of *T. castaneum*.

In summary, the contact toxicity of *E. ciliata* essential oil and its components against adult *T. castaneum* was significantly stronger than that against larvae. A pertinent point in this case is the completion of *T. castaneum* metamorphosis. The adults and larvae of *T. castaneum* are very different [49], especially in terms of self-protection mechanisms and body substances, such as the numerous enzymes that contribute to different degrees of tolerance to external stimuli. In addition, Liang et al. [50] also proved that these two forms differ greatly in their responses to various substances. As described in the literature, the main constituents of the defensive secretions of *T. castaneum* are methyl quinone, 1-pentadecene, 1,6-heptadecadiene, and paeonol. These compounds are repellent to adults whilst being attractive to larvae. Moreover, older adults are more sensitive to these compounds than young adults. Therefore, the whole process of metamorphosis diversifies the response to specific substances, which in turn leads to *E. ciliata* essential oil or its components having dramatically different contact activity against *T. castaneum* adults and larvae. In addition, according to the literature, monoterpenoids and sesquiterpenoid constituents are fast-acting neurotoxins in insects [18]. Both carvone and limonene are monoterpenoids, so it is speculated that carvone and limonene act as fast-acting neurotoxins on pests. In future research, the fumigating mechanism of carvone and limonene will be further explored. In addition, we shoule consider bioactive confrontation of high elsholtzia ketone or dehydroelsholtzia ketone *Elsholtzia* oils with those containing mostly carvone. We also need to consider chiral GC of oil and completion of R- and S-carvone together with R- and S-limonene to use in insect assays.

### 3.4. Carvone Mixed with Limonene and Its Contact Toxicity against T. castaneum Adult

After mixing carvone and limonene in seven different ratios, as shown in Table 4, we found that when the volume ratio of carvone to limonene was 2:6, the CTC value was 134.33, suggesting a synergistic effect (≥ 120). On the other hand, when the volume ratio was 1:7, the CTC showed an additive effect (between 80 and 120). In other ratios, the respective CTCs were less than 80, suggesting an antagonistic effect. As shown in the results, the effect of the limonene mixture appeared unsatisfactory. One of the possible reasons may be that carvone and limonene work in a similar manner; as a result, the addition of limonene inhibits the contact toxicity effect of carvone. The proportion of carvone in *E. ciliata* essential oil was 1.67 times higher than that of limonene, which was equivalent to a compounding agent having a volume ratio of 5:3; the CTC was 67.43 (less than 80), indicating an antagonistic effect. This indicates that the contact toxicity of *E. ciliata* essential oil against *T. castaneum* adults may not be as strong as the contact activity of carvone.

The contact toxicity of carvone against *T. castaneum* larvae displayed enhanced activity by combining in different ratios with limonene. As shown in Table 5, three of the seven ratios had CTC greater than 120 (synergism); these were 1:7, 2:6, and 7:1. In particular, carvone in a 1:7 ratio combination with limonene showed a significant increase in its activity over a single compound with a CTC value of 155. This combination provided strong contact toxicity with the corresponding LD_50_ of 30.04 μg/larva after 24 h of incubation. Besides, when carvone and limonene were mixed in volume ratios of 2:6 and 7:1, the CTC values were 144.08 and 130.19, respectively. The CTCs of these effective combinations were all more than 120, suggesting a synergistic effect. However, when carvone and limonene were mixed in a ratio of 5:3, the CTCs were less than 80, with an antagonistic effect. The 5:3 ratio is similar to the carvone and limonene content ratio in essential oils. Essential oils have stronger contact toxicity against larvae than carvone and limonene, which appears to be a result of synergy among various constituents.

Table 6 shows the fumigation activity of carvone and limonene mixed in different ratios against the adult stage of *T. castaneum*. Out of these seven different ratios, the CTC of two particular ratios were greater than 120 (CTCs of 212.71 and 159.03), suggesting different degrees of synergism. Carvone + limonene at 1:7 ratio combination was found to be most effective in terms of fumigant toxicity against *T. castaneum* adults. This ratio provided strong fumigation activity with corresponding LC_50_ of 2.51 mg/L air after 24 h of incubation. The CTC values of the other ratios of carvone and limonene were less than 80, showing an obvious antagonistic effect.

After the carvone and limonene were mixed in different ratios, the fumigation activity and CTC of the larvae of *T. castaneum* were determined (Table 7). The mixtures of carvone and limonene at 5:3 ratio showed fumigant activity against adult *T. castuneum* (LC_50_ = 20.58 mg/L air). Its CTC value was 89.65, and it appeared to show an additive effect. The values of CTC under other ratios were all less than 80 and thus suggested an antagonistic effect.

Figure 1 shows a general synergistic effect and antagonistic effect (to some degree) with different mixture ratios in terms of contact toxicity against the adult and larval stages of *T. castaneum*. The figure also indicates a deviation in the CTC value trends between adult and larval stages. We observed that when the mixture ratio was 2:6, the CTC values for both stages were greater than 120, which suggested synergism, particularly in larvae. The CTC value reached the maximum when carvone and limonene were mixed at a ratio of 1:7. This result implies that synergy for larvae is the best at a 1:7 ratio. However, at this ratio, the CTC value of adults was 85.97, indicating an additive effect. In addition, when the mixture ratios were 3:5 and 4:4, the CTC value of the contact killing effect in adult and larval stages decreased significantly. The CTC of larvae showed an upward trend after 4:4, reaching 130.19 at a ratio of 1:7, indicating a synergistic effect. On the contrary, the effect on adult *T. castaneum* declined after the mixture ratio of 4:4 and reached 1.01 at the ratio of 7:1, indicating a marked antagonistic effect. In conclusion, except at the ratio of 4:4, the CTC values of the *T. castaneum* larvae were slightly higher than those of adults in the same ratios. It can be deducted that the contact toxicity effect of carvone and limonene on the larvae of *T. castaneum* is generally better than that of adults at the same ratio.

When carvone and limonene were mixed in different ratios, we observed obvious differences in fumigation activity against the adult and larval stages of *T. castaneum* (Figure 2). The CTC of adults reached the maximum value with the best synergistic effect at a ratio of 1:7. Moreover, the mixture showed a synergistic effect when the ratio was 2:6. After that, the value of CTC was less than 80, which indicated an antagonistic effect. However, the co-toxic effect against the larvae was generally weak or appeared antagonistic, except when the ratio was 5:3, which showed an additive effect. Generally, the CTC values of *T. castaneum* adults were slightly higher than the larvae using the same mixture ratio. Therefore, when carvone and limonene were mixed in the same ratio, its fumigation activity is better in adults than in larvae.

Through the mixture of the two major components, we identified the optimal mixing method that can effectively target *T. castaneum*. Changing the mixing ratio also changed the insecticide effects of the two compounds, but the effect of getting twice the result with half the effort was achieved for both plant essential oil mixed with compounds as well as essential oils mixed with essential oils. For example, the Commonwealth Scientific and Industrial Research Organisation (CSIRO) Institute of Entomology compared several natural plant extracts known to have insecticidal activity with ethyl formate and found that some plant products have a good synergistic effect [52]. The essential oils from plants have the advantages of having broad-spectrum insecticidal efficacy and being generally safe in humans, animals, and the environment. Carvone and limonene are derived from plant essential oil with the synergistic effect produced at a volume ratio of 2:6. The difference in the mode of action of the two substances against *T. castaneum* are important factors that influence its compounding effect. Exploring ways to make better use of mixed medicines will not only help overcome the high cost of plant essential oils but will also provide a theoretical basis for the practical application of the two medicines.

## 4. Conclusions

In this study, nine different components were identified from *E. ciliata* essential oil extract. The two main components, carvone and limonene, showed strong contact and fumigation activities against adults and larvae of *T. castaneum*. Meanwhile, *E. ciliata* essential oil also showed intense toxicity against the test insects. We also found that carvone might play a key role in the contact toxicity of *E. ciliata* essential oil against *T. castaneum*. Carvone and limonene exhibited synergistic effects at a volume ratio of 2:6. Altogether, our results suggest that *E. ciliata* essential oil extract and its two major components have a potential for downstream development as natural insecticides.

## Figures and Tables

**Figure 1 foods-09-00345-f001:**
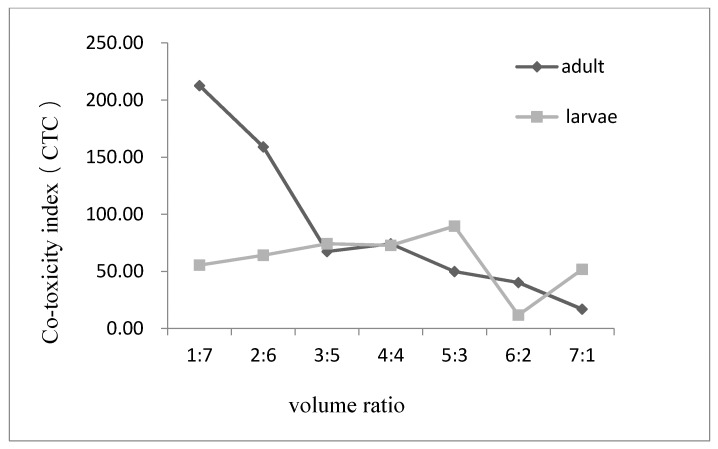
The CTC of contact activity of carvone and limonene at different ratios against adults and larvae of *T. castaneum*.

**Figure 2 foods-09-00345-f002:**
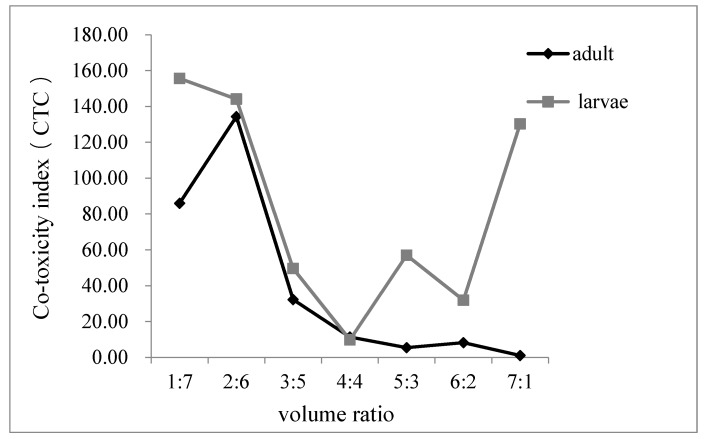
The CTC of fumigant activity of carvone and limonene at different ratios against adults and larvae of *T. castaneum*.

**Table 1 foods-09-00345-t001:** Chemical composition of the essential oil from *E. ciliata*.

Number	Constituent	Retention Time/Min (Rt)	Ri *	Relative Content (%)
1	*α*-Pinene	3.394	932	0.55
2	*β*-Pinene	3.812	977	0.74
3	Myrcene	3.861	988	1.02
4	*β*-Phellandrene	3.966	1019	0.19
5	Limonene	4.464	1040	22.05
6	*β*-Ocimene	4.654	1061	4.08
7	Linalool	5.367	1090	0.83
8	Elsholtzia ketone	6.726	1199	1.02
9	Carvone	7.366	1216	31.63
10	Dehydroelsholtzia ketone	8.104	1277	14.86
11	Cubebene	9.180	1344	1.06
12	*β*-Bourbonene	9.635	1379	0.44
13	*β*-Caryophyllen	10.077	1414	2.92
14	*α*-Caryophyllene	10.397	1450	15.47
15	(-)-Humulene epoxide II	10.643	1454	0.25
16	α-Farnesene	11.965	1489	0.47
-	Total	-	-	97.58
	Others			2.42

* RI (retention index) as determined on a HP-5MS column using the homologous series of *n*-hydrocarbons.

**Table 2 foods-09-00345-t002:** Contact toxicity of *E. ciliata* essential oil and its main constituents against *T. castaneum*.

*T. castaneum*	Treatment	Ld_50_ (mg/Adult)	95% Fl (mg/Adult)	Slope ± Se	*p*-Value	Chi Square X^2^
Adult	Essential oil	7.79	6.96−8.65	4.14 ± 0.46	0.85	16.17
	Carvone	5.08	4.19−6.20	4.30 ± 0.46	0.01	44.15
	Limonene	38.57	34.48−43.09	3.84 ± 0.42	0.55	21.54
	Pyrethrin	0.09	0.08−0.11	2.48 ± 0.31	0.92	14.27
Larva	Essential oil	24.87	19.55−30.69	1.69 ± 0.22	0.64	24.72
	Carvone	33.03	26.55−41.26	1.86 ± 0.23	0.75	18.12
	Limonene	49.68	34.10−84.04	0.95 ± 0.15	0.54	26.70
	Pyrethrin	1.31	0.75−2.17	0.80 ± 0.10	0.82	16.72

**Table 3 foods-09-00345-t003:** Fumigant toxicity of *E. ciliata* essential oil and its main constituents against *T. castaneum*.

*T. castaneum*	Treatment	LC_50_ ( mg/L Air)	95% FL (mg/L Air)	Slope ± SE	*p*-Value	Chi Square X^2^
Adult	Essential oil	11.61	9.21−14.01	4.39 ± 0.47	0.00	87.62
	Carvone	4.34	3.90−4.84	6.27 ± 0.83	0.98	7.89
	Limonene	5.52	2.75−9.22	1.69 ± 0.47	0.83	5.85
	Methyl bromide ^a^	1.83	1.43−2.23	4.90 ± 0.51	0.89	8.67
Larva	Essential oil	8.73	6.62−11.25	1.42 ± 0.17	0.99	11.19
	Carvone	28.71	23.07−36.05	1.63 ± 0.15	0.36	35.41
	Limonene	20.64	16.96−25.56	1.71 ± 0.16	0.86	24.46
	Phoxim	1.05	1.23–2.08	1.65 ± 0.45	0.89	3.25

^a^ The data for methyl bromide was derived from the literature with a consistent experimental method [51].

**Table 4 foods-09-00345-t004:** Contact toxicity and CTC () of carvone and limonene mixture against *T. castaneum* adults.

Volume Ratio	LD_50_ (μg/Adult)	Slope ± SE	*p*-Value	ATI	TTI	CTC
1:7	24.60	2.935 ± 0.59	0.64	20.65	24.02	85.97
2:6	10.84	2.972 ± 0.51	0.54	46.85	34.88	134.33
3:5	34.43	1.856 ± 0.45	0.99	14.75	45.73	32.26
4:4	39.60	1.970 ± 0.47	0.99	12.83	113.17	11.33
5:3	140.30	1.605 ± 0.79	0.60	3.62	67.43	5.37
6:2	79.34	1.666 ± 0.95	0.95	6.40	78.29	8.18
7:1	434.82	1.495 ± 0.85	0.80	1.17	115.93	1.01

**Table 5 foods-09-00345-t005:** Contact toxicity and CTC of carvone and limonene mixture against larvae of *T. castaneum*.

Volume Ratio	LD_50_ (μg/Larva)	Slope ± SE	*p*-Value	ATI	TTI	CTC
1:7	30.04	2.323 ± 0.59	0.34	109.94	70.68	155.55
2:6	30.62	3.829 ± 0.73	0.71	107.87	74.87	144.08
3:5	84.30	1.145 ± 0.14	0.88	39.18	79.06	49.56
4:4	405.96	1.390 ± 0.12	0.46	8.14	83.24	9.77
5:3	66.30	3.074 ± 0.16	0.95	49.82	87.43	56.98
6:2	112.98	2.175 ± 0.94	0.90	29.24	91.62	31.91
7:1	26.48	5.321 ± 0.11	0.96	124.74	95.81	130.19

**Table 6 foods-09-00345-t006:** Fumigant toxicity and CTC of carvone and limonene mixture against adult of *T. castaneum*.

Volume Ratio	LC_50_ (mg/L Air)	Slope ± SE	*p*-Value	ATI	TTI	CTC
1:7	2.51	2.921 ± 0.48	0.00	172.91	81.29	212.71
2:6	3.25	4.793 ± 0.63	0.05	133.54	83.97	159.03
3:5	7.43	2.845 ± 0.93	0.80	58.41	86.64	67.42
4:4	6.55	2.656 ± 0.82	0.72	66.26	89.31	74.19
5:3	9.45	2.567 ± 0.76	0.88	45.93	91.98	49.93
6:2	11.39	2.814 ± 0.44	0.85	38.10	94.66	40.25
7:1	26.30	1.889 ± 0.40	0.72	16.50	97.33	16.95

**Table 7 foods-09-00345-t007:** Fumigant toxicity and CTC of carvone and limonene mixture against larvae of *T. castaneum*.

Volume Ratio	LC_50_ (mg/L Air)	Slope ± SE	*p*-Value	ATI	TTI	CTC
1:7	38.56	1.846 ± 0.70	0.87	74.46	134.21	55.48
2:6	34.64	2.314 ± 0.77	0.80	82.88	129.32	64.09
3:5	31.08	2.201 ± 0.71	0.88	92.37	124.44	74.23
4:4	32.99	2.286 ± 0.72	0.94	87.03	119.55	72.79
5:3	27.93	3.041 ± 0.79	0.54	102.79	114.66	89.65
6:2	221.59	1.215 ± 0.85	0.81	12.96	109.76	11.80
7:1	52.91	3.189 ± 0.46	0.88	54.26	104.89	51.73

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
