# Peer review of "Toxicity and Synergistic Effect of Elsholtzia ciliata Essential Oil and Its Main Components against the Adult and Larval Stages of Tribolium castaneum"

_foods, 2020, doi:10.3390/foods9030345_

Round 1

Reviewer 1 Report

The article entitled “Toxicity and Synergistic Effect of Elsholtzia ciliata Essential Oil Extract and Its Main Components against the Adult and Larval Stages of Tribolium castaneum” authored by Jun-yu Liang 1, Jie Xu, Ying-ying Yang, Ya-zhou Shao, Feng Zhou and Jun-long Wang presents a thorough study on the effect of essential oil derived from Elsholtzia ciliate and its components on the pest/insect control experiments. The study presents many interesting points for the scientific community involved/interested in the research on natural phytochemicals alternatives to synthetic chemicals; however, a few more points are required to be addressed/ added. The points are as follows.

  1. Lines 99-100, 110-111; Experiments were repeated for 5 times (concentration), the standard deviation is a missing part in the data presented in the tables and the graphs.
  2. Lines 135-147: In this study the main compounds discussed are limonene and carvone. The yields of these compounds in Elsholtzia ciliate collected from the different regions in China have been observed to be different. The paragraph describes about the yield comparison of different compounds other than those studied in this work. The authors are requested to present the relevant information in this regard.
  3. The working mechanism of toxicity and fumigation processes on larvae and adult insects are different. The authors are requested to add a brief description of the mode of action of limonene and carvone in the discussion section.
  4. The article is well structured and nicely describing the study carried out. I recommend acceptance after incorporation of a few modifications suggested above.

Author Response

Dear Reviewer:

     Thank you for your useful comments and suggestions on the language and structure of our manuscript. We have modified the manuscript accordingly and detailed corrections are listed  point by point. Please see the attachment and manuscript for details. 

Sincerely,

Jun Yu Liang and Jie Xu on behalf of the authors.

Reviewer 2 Report

In this manuscript Jun-yu Liang and colleagues have analyzed the toxicity and synergistic effect of Elsholtzia ciliata essential oil extract and its main components  against the adult and larval stages of Tribolium castaneum. The authors utilized appropriate technique of characterization and analysis, supplying useful information and,  overall, the work shows clear results with a detail analysis of literature data. The work is well written and organized, there are only some typing errors to correct during the revision of the work. Moreover some sentences are too complex and rambling and, in my opinion, is useful to further fortify the “Conclusions” section to better explain its potentiality.

Author Response

(The authors gave the same response as above.)

Reviewer 3 Report

== MAIN ==
1
What is the advantage of Elshscholtzia ciliata essential oil (ECEO) over other oils containing phenol (e.g. carvone) and hydrocarbon (e.g. limonene)?
Is E.c. a plant of a particular oil yield? -> 0.3% = No
-> Think on the readers' judgment and supply the INTRO. Also, see the comments below.

INTRO
Are carvone or limonene not toxic to the humans? What are their effects on the skin, whole body and so on?
-> Supply the INTRO.

l.61, CLAIMS, INTRO
'However, literature surveys have shown no report on chemical composition and insecticidal activity of the essential oil of E. ciliate aerial parts against T. castaneum. The present study was therefore undertaken to investigate the chemical components and insecticidal activities of the essential oil, including its active biochemical constituents against the food storage pest T. castaneum.'

There are hundreds of works in this area.
-> At least, the following literature should be cited in INTRO part :
- works describing the action of carvone and its analogs on such insects,
- works describing the action of limonene and its analogs on such insects,
- works describing the action of alpha-caryophyllene and its analogs on such insects,
- work describing essential oil chemotypes of Elsholtzia ciliata - to enter the readers into the subject and to inform on possible differences in activity determined for the ECEO (present in part in RESULTS),
- works describing cost/effect calculation when ANY essential oils are used as insect deterrents or poisons.

l.58
-> The cited concentrations should be critically interpreted, e.g., '145.5 mg/cm2' is relatively high concentration - equivalent to ~100µl/cm2 - Is it applicable at all?

l.98, 109
CHEMICALS subchapter is missing. Such info should be given there: who was a supplier of, e.g., pyrethrin and what was declared purity of this standard (and each other)?
-> Supply.
-> Supplier, purity and stereoisomer of carvone and limonene used for activity testing must be given.

l.145-147
Composition-related works given in RESULTS are partial.
Observation on different compositions of ECEO reported in the literature is given but the conclusion for readers is lacking.
-> See far above. Supply.

l. 163-165, 177-179, and elsewhere
Exaggerated, in my opinion.
-> Justify, using economic calculations. Consider:
-> What is the effective deterrent / inhibiting concentration of oil in the atmosphere -> [TIME] of fumigation needed with [AMOUNT] of ECEO per [e.g., SQUARE METER] of a protected area. -> That means [AMOUNT+COST] of oil per standard storage. The current cost is as follows [COST ECEO and active compounds in the best ratio].

EXPERIMENTAL

Was it checked by chiral GC if commercial carvone was the same stereoisomer as in essential oil, and similarly limonene? Several general works describe stereoisomer-insect response relations.
-> Supply.

RESULTS
-> Lack of comparison and comments on relative ratio of compounds in tested ECEO (3:2) to analogous CTC. Should be added to the charts.
-> Ratios of, e.g., studied Contact activity that are ~10^2-fold weaker than positive control should be commented.

== OTHERS ==
l.13, 33
'Recent investigations indicated that insects pose a new risk for global food security by damaging food crops and other food products derived from plants'
Are the pests a 'new risk'? What 'recent' is there? Did ancient or primitive peoples have the same problem, didn't they? What about an old idea to synthesize pyrethroids or DDT-like substances (the last honored with Nobel Prize already in 1948)?
-> Think on the judgment and correct the INTRO.

l.20-25
Are the LD50 given for Contact Toxicity and LC50 for Fumigant Activity in RESULTS? Why are other units presented here?
-> Unify.

l.53
'These essential oils can be degraded easily by detoxifying enzymes and the products are also not risky for other organisms'
If insects possess appropriate enzymes, why bother them with essential oils?
-> Precise the subject of this sentence or supply INTRO.

l.68
The coordinates point a place in Algeria, not in China. -> Correct.
Who was responsible for botanical plant identification. - > Supply.

l.71
Was this a steam distillation or water distillation? There is a difference. -> Clear.

l.95, 106
What gradients? Where can we found them? -> Supply.

l.107
How can we measure the area in cm? -> Correct.

l.128-132
Check the integrity of the presented equations with FOODS manuscript template.

l.135
'v' is for velocity in IS, use 'V' for volume.
'w' is for weight in IS, use 'm' for mass

== LANGUAGE ==
l.3, 27
What kind of 'extract' was analyzed? We can fancifully name the 'essential oil' as an extract but who do this? -> Think about the more accurate term.

l.19
Are carvone or limonene 'species'?

l.55
'Elsholtzia ciliate'
-> First mention in the text should be in full (with botanical author citation, and family)
-> and without TYPO, that is continued in the text

l.59
'Elsholtzia stauntoni benth' no need to cite botanical author, moreover with a TYPO

l.73
'extracted rate' means?

l.76
What is yeast flour?

========
Strengths of this work are:
experiment with different carvone to limonene ratios.
-> Try to improve all the work and highlight the present novelty. If you don't want to encourage the readers to use pure chemicals - consider listing of other plants that more efficiently yield the essential oils with mentioned compounds. A contact with a specialist in essential oils area will be valuable if you cannot do this by yourself.

Author Response

(The authors gave the same response as above.)

Round 2

Reviewer 3 Report

Dear Authors,

Thanks for improveing of your manuscript.
It looks almost done.

1
Sorry for 'phenol (e.g. carvone)' in my last review. My obvious fault.

2
I have asked to remove botanical typos, but are still present. I understand language differences but Authors are obliged to use proper names.

l.28, 66, 71, 173, 417 (probably a reference is a source of typo)
'ciliate' -> 'ciliata'
l.200
'Ciliata' -> 'ciliata'
l.68
'stauntoni' -> 'stauntonii'

3
A short survey of English language corrector is needed (to point out and correct all minor failings of manuscript).
e.g. l.51 'ies'; l.59 'Researchs'; 'And' as first word in some sentences -> To be rewritten.

4
l.95-97
Response 14: It was the steam distillation, which is clear in the manuscript.
-> No, it isn't. Thus I have asked.

Two contradictory statements in one paragraph:
'A total of 500 g minced sample was placed into a round bottom flask with 2000 mL water added.' - suggests water distillation (where plant material is mixed with water and boiled in one bulb/container to distill essential oil; that was probably done, because it is much easier in lab-scale)
'To obtain the crude essential oil, the minced plant was connected to the steam distillation unit and maintained for 6 h.' - suggests steam distillation (where plant material is initially dry; steam from a separate, steam-producing unit, takes the essential oil from dry plant material). Condenser or Clevenger-type apparatus is not a 'steam distilaltion unit'. -> Just remove 'steam'.
If you still have doubts, refer to any serious organic chemistry manual. In poor works pop-mistakes exists (and are perpetuated). Name properly what was done.

l.160
If so, the purity level (27%) disqualify the pyrethrins as an analytical standard, sorry.

l.166
I asked to change 'w' to 'm' because you don't measure the 'weight' but the 'mass' in this case. Basic, but usually neglected. - > To be corrected.

l.169
Is carvone a ketone or not? If so, 31.6+14.9+1.0 is not 15.9.
Are all listed compounds terpenoids or not? If so, 31+22+15+14 is not 76%.
-> Correct both. If you distinguish terpenes (hydrocarbons) and terpenoids (oxygenated forms) -> Clear it.
-> BTW, avoid unnecessary digits (your measurement looks like GC single-hit, not replicated; e.g. 0.8 is not worse than 0.83)

l.245
'phoxim' is still not listed in Chemicals -> Supply.

l.214-215
In your future research,
- Consider confrontation of high Els.-ketone or dihydroEls.-ketone Elsholtzia oils with those containing mostly carvone. (Aldehydes and) ketones are usually bioactive.
- Consider chiral GC of oil and completion of R- and S-carvone together with R- and S-limonene to use in insect assays (all are cheap and available with high purity; results may be completely different).

Good luck.

Author Response

Dear Reviewer:

     Thank you for your useful comments and suggestions of our manuscript. We have modified the manuscript accordingly and detailed corrections are listed  point by point. Please see the attachment and manuscript for details. 

Sincerely,

Jun Yu Liang and Jie Xu on behalf of the authors.
